# The Glycan Ectodomain of SARS-CoV-2 Spike Protein Modulates Cytokine Production and Expression of CD206 Mannose Receptor in PBMC Cultures of Pre-COVID-19 Healthy Subjects

**DOI:** 10.3390/v16040497

**Published:** 2024-03-24

**Authors:** Cristiana Barbati, Carla Bromuro, Silvia Vendetti, Antonella Torosantucci, Roberto Cauda, Antonio Cassone, Carla Palma

**Affiliations:** 1Department of Infectious Diseases, Istituto Superiore di Sanità, Viale Regina Elena, 299, 00161 Rome, Italy; cristiana.barbati@iss.it (C.B.); carla.bromuro@iss.it (C.B.); silvia.vendetti@iss.it (S.V.); a.torosantucci53@gmail.com (A.T.); 2Dipartimento Salute e Bioetica, Sezione Malattie Infettive, Policlinico Universitario A. Gemelli IRCCS, Largo Agostino Gemelli, 8, 00136 Rome, Italy; roberto.cauda@unicatt.it; 3Polo d’Innovazione della Genomica, Genetica e Biologia, Strada del Petriccio e Belriguardo 35, 53100 Siena, Italy

**Keywords:** SARS-CoV-2 spike proteins, PBMC, immunomodulation, IL-6, IFN-γ, T cell mitogen, mannose receptor (CD206)

## Abstract

The ability of recombinant, SARS-CoV-2 Spike (S) protein to modulate the production of two COVID-19 relevant, pro-inflammatory cytokines (IL-6 and IFN-γ) in PBMC cultures of healthy, pre-COVID-19 subjects was investigated. We observed that cytokine production was largely and diversely modulated by the S protein depending on antigen or mitogen stimulation, as well as on the protein source, insect (S-in) or human (S-hu) cells. While both proteins co-stimulated cytokine production by polyclonally CD3-activated T cells, PBMC activation by the mitogenic lectin Concanavalin A (Con A) was up-modulated by S-hu protein and down-modulated by S-in protein. These modulatory effects were likely mediated by the S glycans, as demonstrated by direct Con A-S binding experiments and use of yeast mannan as Con A binder. While being ineffective in modulating memory antigenic T cell responses, the S proteins and mannan were able to induce IL-6 production in unstimulated PBMC cultures and upregulate the expression of the mannose receptor (CD206), a marker of anti-inflammatory M2 macrophage. Our data point to a relevant role of N-glycans, particularly N-mannosidic chains, decorating the S protein in the immunomodulatory effects here reported. These novel biological activities of the S glycan ectodomain may add to the comprehension of COVID-19 pathology and immunity to SARS-CoV-2.

## 1. Introduction

The Severe Acute Respiratory Syndrome–Coronavirus-2 (SARS-CoV-2) is the aetiologic agent of COVID-19, a pandemic disease, which started in China at the end of 2019, spread worldwide and caused, as of last December, nearly two billion infections and seven million deaths, mostly in elderly subjects with one or more comorbidities [1]. At least as many deaths as above have been avoided using highly effective SARS-CoV-2 vaccines, which have been generated at an unprecedented speed, made available worldwide, but unequally distributed. The formulation of most of these vaccines has been based on the Spike (S) constituent of the virus, or its coding DNA or RNA sequences. The S protein is an immunodominant antigen constituted by a trimeric glycoprotein, each monomeric unit of which has two subunits: S1 and S2, mediating binding of the virus to host cell surface (S1) and fusing with the cell membrane (S2) [2].

The S protein plays an essential role in infection and disease. Through its receptor-binding domain, the S protein binds to the human receptor ACE2, a critical component of blood pressure homeostasis, and allows the virus to enter the cells [3]. There is ample evidence that inflammation concurs with virus reproduction in the pathogenesis of the disease, particularly its severe form such as the CRS (cytokine release syndrome) [4,5]. Coherently, it has been shown that inflammation control by pharmacological or immunological interventions greatly benefits the infected patients [6,7]. Multiple virus constituents can potentially induce inflammatory response, and there is some evidence about the role of the S protein itself as inflammation inducer [5,6,7,8,9,10]. However, mechanisms and molecular determinants of inflammation caused by this glycoprotein are still poorly known.

The N-linked glycans of glycoproteins activate and modulate adaptive and innate responses by interacting with C-type lectin receptors expressed on both lymphocytes and phagocytic or antigen-presenting cells [11]. The SARS-CoV-2 S protein is extensively decorated by various types of glycans that appear to be substantially conserved even in distant SARS-CoV-2 variants [9]. Actually, there are 22 N-glycan sites present on each monomeric S protein subunit and bearing N-glycans of different composition [12,13]. It has been reported that circulating mannose-binding lectin (MBL), a collectin acting as pattern recognition in the first line of defense in the pre-immune host, specifically interacts with the active trimer of SARS-CoV-2 S protein through the carbohydrate region domain [14]. MBL-interaction with the S protein activates complement through the lectin pathway, which was considered to be relevant in controlling COVID-19 disease. An association between MBL2 haplotypes or biallelic variants and COVID-19 severity in human patients has been reported [14].

Considering the number and complexity of receptor/ligand systems whose interactions depend on the glycan moieties and their relevance in anti-pathogen immunity [11,15,16], the biological significance of glycan moieties decorating the S protein of SARS-CoV-2 is an attractive topic for research aimed at ascertaining the role of N-glycans in protection from, and/or pathology of, COVID-19.

Along this line of research, we have investigated the immune-modifying properties of two recombinant preparations of the S protein, generated in human Hek293 cells (S-hu) or insect cells (S-in). We focused on S protein capacity for modulating the production of two COVID-19-relevant, pro-inflammatory cytokines such as Interleukin-6 (IL-6) and Interferon-γ (IFN-γ) [4,5] in both primary and secondary immune responses, including those to polyclonal, mitogenic activators. To this end, we used peripheral blood mononuclear cells (PBMC) from healthy donors whose blood was collected in the pre-pandemic era in order to avoid possible interference of immunity to S protein due to infection or vaccination.

## 2. Materials and Methods

### 2.1. SARS-CoV-2 S Proteins

Recombinant Sar-Cov-2-S proteins were purchased from SinoBiological, Beijing, China: Sar-Cov-2 (2019-nCoV) Spike (S1+S2) ECD-His recombinant protein produced in baculovirus-insect cells 1209 AA MW 134.36 KDa, binding ability to human ACE2 in a functional ELISA EC50 = 400–1200 ng/mL (cod. 40589-V08B1); Sar-Cov-2 (2019-nCoV) Spike (S1+S2) ECD (R683A, R685A, F817P, A892P, A942P, K986P, V987P)-His recombinant protein produced in human HEK293 cells 1248 AA MW 138.5 KDa, binding ability to human ACE2 in a functional ELISA EC50 = 10–50 ng/mL (cod. 40589-V08H4); Sar-Cov-2 (2019-nCoV) Spike S1-His recombinant protein produced in human HEK293 cells 681AA MW 76.5 KDa, binding ability to human ACE2 in a functional ELISA EC50 = 200–600 ng/mL (cod. 40589-V08H); and Sar-Cov-2 (2019-nCoV) Spike S2 ECD(708S-1207E)-His recombinant protein produced in human HEK293 cells 511 AA MW 56.3 KDa, inability to bind human ACE2 in a functional ELISA (cod. 40589-V08H1).

### 2.2. Binding Assay of S Proteins to Concanavalin A

Binding of S proteins to Concanavalin A (Con A) was measured by an in-house ELISA assay. Briefly, polystyrene 96-well plates (Nunc, Thermo Fisher Scientific, Herndon, VA, USA) were coated and incubated overnight at 4 °C with 100 μL/well of 0.75 μg/mL of S-in, S-hu, and S1 and S2 subunits in PBS. Coated plates were then washed 4 times with PBS containing 0.05% Tween 20 (PBS-T). Plates were blocked for 1 h at 37 °C with 3% BSA (Sigma-Aldrich, St. Louis, MO, USA) in PBS-T. After the BSA solution was removed, in a first set of experiments, the wells of plates coated with S-in or S-hu were incubated for 1 h at 37 °C with 0.0125 μg/mL of HRP-conjugated Con A (Sigma L-6397) in PBS-T containing 3% BSA, alone or in combination with scalar concentrations of mannan (*S. cerevisiae*, Sigma M3640); dextran 1000 (Fluca 31416); alfa-methyl-mannoside (Sigma-Aldrich, REF. M6882); N-acetil-glucosamine (Sigma-Aldrich, REF. A8625). Con A and oligosaccharides were simultaneously added to the plates. In a second set of experiments, wells of plates coated with S-in, S-hu, S1 or S2 (each at 0.75 μg/mL) were incubated for 1 h at 37 °C with 0.025 ug/mL of HRP-conjugated Con A in PBS-T containing 3% BSA, alone or in combination with 50 μg/mL mannan added simultaneously to the plates. The plates were then washed 3 times and incubated with TMB-ELISA (Thermo Fisher Scientific, REF.34028) for 30 min at RT. Color development was stopped with 2 M H_2_SO_4_. The binding of Con A to S proteins was revealed by measuring absorbance at 450 nm in a spectrophotometer (Multiskan FC, Microplate Reader REF. 51119000, Thermo Fisher Scientific). Data were presented as the mean of optical density (OD) corrected for background (wells without coated antigen).

### 2.3. Human PBMC and Cell Cultures

Human studies were performed in accordance with the ethical guidelines of the 1975 Declaration of Helsinki. PBMC were isolated from buffy coat samples of healthy individuals attending the transfusion center of the bloodbank of the University “La Sapienza”, Rome, Italy in the years 2016–2017, using density centrifugation. Cells were resuspended in foetal bovine serum (FBS) (Euroclone, Pero, IT) containing 10% dimethylsufoxide (Sigma-Aldrich) and aliquoted at an approximate concentration of 10–20 × 10^6^ cells per cryovial, frozen overnight at −80 °C and transferred to liquid nitrogen storage the following day. Cryopreserved PBMC were thawed in a water bath at 37 °C, washed in warmed PBS, resuspended at an approximate concentration of 2–3 × 10^6^ cells per ml of RPMI-1640 (Sigma life science) containing 10% heat-inactivated human AB serum, 2 mM l-glutamine 10 mM HEPES buffer and 50 U/mL penicillin and 50 μg/mL streptomycin (complete RPMI-medium), and added with 2 μL/mL of 25 U benzonase (Sigma-Aldrich REF. E1014) (final concentration 50 U/mL). Cells were rested at 37 °C for 2 h with 5% CO_2_, washed and then cultured. After counting, PBMC were cultured at a density of 2 × 10^6^ cells/mL in 96-well plates (final volume 250 µL) or in 48-well plates (0.8–1 mL) in complete RPMI-1640-medium and treated for 72 h with the following stimuli: ConA (Sigma-Aldrich, REF C-0412) at 4 µg/mL; or agonist anti-CD3 monoclonal antibody (mAb) (clone UCHT1, Invitrogen Life Technologies, Carlsbad, CA, USA) at 0.8 µg/mL; or full-length non-glycosylated recombinant protein portion of cell surface mannoprotein MP65 (MP65) from *Candida albicans* (GenScript, Piscataway, NJ, USA) at 1 µg/mL. All stimulations were performed alone or in combination with S- proteins: S-insect (S-in), or S-human HEK923 (S-hu), or S1 or S2 subunits, each at 5 µg/mL, or mannan from *Saccharomyces cerevisiae* at 10 µg/mL. In some cultures, combinations of mannan with S proteins in the presence or absence of anti-CD3 mAb were also performed. The cells were also cultured with only spike proteins or mannan. At the end of the cultures, cell culture supernatants were collected, spun free from cell and debris by centrifugation (2000 rpm for 7 min), and used to detect cytokine production. Cells were used to measure vitality by XTT assay, or CD4 and CD8 T cell proliferation by CFSE assay or expression of mannose receptor on macrophages by flow cytometry.

### 2.4. IFN-γ and IL-6 Quantification

Cytokines were detected in supernatants of cultured PBMC by quantitative sandwich ELISA specific for IFN-γ, or IL-6 (human Quantikine, R&D Systems, Inc. Minneapolis, MN, USA), in accordance with manufacturer’s instructions. Quantifications were all calculated using a standard curve obtained for each cytokine standard provided by the manufacturer.

### 2.5. In Vitro Toxicology Assay XTT-Based

For XTT assay, PBMC were cultured in complete RPMI medium using an RPMI-1640 without phenol red (Gibco, Thermo Fisher Scientific) in 96-well plates, as described above. After 96 h of culture, 50 μL of medium were removed and substituted with fresh RPMI without phenol red containing XTT 1 mg/mL (Sigma-Aldrich, in vitro toxicology assay kit XTT-based, TOX-2) to reach an amount equal to 20% of the culture medium volume. Cells were incubated for a further 6 h before measuring absorbance at a wavelength of 450 nm. The OD values of the XTT test measure the activity of cells via mitochondrial dehydrogenases and depend on the number and metabolic activity of living cells.

### 2.6. CFSE Assay

PBMC (10^7^ cells/mL) were stained with CFSE (Invitrogen Life Technologies) at 1 μM in PBS 1%FBS for 10 min at 37 °C in the dark, then washed and cultured for 3 days (as described above). After culture, cells were washed with FACS buffer and stained for 20 min at 4 °C with the following monoclonal antibodies (mAb): PE anti-human CD4 (clone RPA-T4, BD Biosciences, San Jose, CA, USA), PerCP anti-human CD8 (clone HIT8A, BD Biosciences) and APC anti-human CD3 (clone MEM-57, Immunological Sciences, Roma, IT) or the isotype controls. After washing, cells were transferred into FACS tubes and analyzed. The acquisition was performed on a FACSCalibur cytometer (BD Immunocytometry Systems, San Jose, CA, USA), and the data were analyzed using CellQuest Pro software (BD Immunocytometry Systems).

### 2.7. Mannose Receptor Expression by Flow Cytometry

PBMC were cultured in the presence or absence of S proteins, mannan or their combination for 72 h in complete RPMI medium. In some experiments, after 72 h of culture, the stimulants were added for the last 30 min of culture before collecting cells for staining. Cultured cells were washed and then stained for 20 min at 4 °C with the following mAbs: PE-Cy5 anti-human CD14 (clone M5E2, BD Biosciences) and APC mouse anti-human CD206 (clone 19.2, BD Pharmingen, San Diego, CA, USA). After washing, cells were transferred into FACS tubes and analyzed. Before surface staining with specific mAb, the Fc receptors were blocked using purified Rat Anti-Mouse CD16/CD32 (Clone 2.4G2, BD Pharmigen) at a concentration of 5 µg/mL for 10 min at 4 °C. Staining of samples with isotype controls was used as a reference to determine positive and negative populations. The acquisition was performed on a FACSCalibur cytometer and the data were analyzed using the Cell Quest Pro software.

### 2.8. Statistical Analyses

Data were analyzed using GraphPad Prism 6.0 software (San Diego, CA, USA). The comparisons between two groups of non-parametric data were analyzed using a Wilcoxon matched-pairs signed rank test, and parametric data were analyzed by a paired *t* test; for multiple comparison, the data were analyzed by one-way ANOVA with Tukey’s multiple comparisons post hoc test. A *p* value < 0.05 was taken as indicative of a statistically significant difference.

## 3. Results

### 3.1. SARS-CoV-2 S Protein Binds Con A through Its Glycan (Mannosides) Constituents

The oligosaccharide constituents of proteins are used by viruses and bacteria to interact with a host immune system [11,15,16]. First, we investigated whether the SARS-CoV-2 S protein, which does indeed contain glycan moieties [12,13], was able to bind to the Con A, a lectin capable of binding glycan (preferably mannan) moieties [17] and known to be a potent lymphocyte mitogen [18].

Using an in-house developed Elisa assay, we found that a HRP-conjugated Con A strongly bound both the S-in and the S-hu proteins, with similar efficiency (Figure 1A,B). The binding of both S-in and S-hu protein to Con A was markedly inhibited by the yeast polysaccharide mannan (Figure 1A). Only a little difference in potency was observed at very low mannan concentrations (0.078 and 0.039 μg/mL) between the two S proteins (Appendix A). Alfa-methyl-mannoside also, in a dose-response manner, and to a lesser extent than mannan, inhibited the S protein-Con A binding without appreciable differences between S-in or S-hu (Figure 1A and Appendix A). As expected, dextran or N-acetyilglucosamine had no relevant inhibitory capacity in the S-protein-Con A binding assay (Figure 1A). A low degree of binding to Con A was also shown by the S1 and S2 subunits of S-hu protein, and this was also inhibited by mannan (Figure 1B). Overall, the data indicated that the trimeric forms of both S proteins, and to a lower degree, also the monomeric subunits S1 and S2 of the S-hu protein, were able to bind the Con A, mainly through their mannan moieties. We therefore verified whether the above interaction could interfere with the mitogenic activity of the lectin.

### 3.2. S-in Protein Prevents Activation of PBMC Stimulated with Con A

To investigate whether and to what extent the S protein of SARS-CoV-2 was able to modulate cytokine production, cryo-preserved PBMC of pre-pandemic healthy humans were used to exclude possible interference by pre-existent antigenic immune responses to the protein. PBMC were stimulated with Con A, a lectin known to activate CD4 and CD8 T lymphocytes [18,19]. In the absence of the S protein, the Con A-stimulated PBMC showed the expected, donor-variable production of IFN-γ, a main mediator of Th1 responses, and IL-6, a cytokine which concurs with inflammation and the dangerous cytokine storm associated with COVID-19 [4,5] (Figure 2A,B). In the presence of S-in protein, the Con A-induced cytokine production was significantly inhibited, though with quantitative differences among the different donors. Of note, the reduction in Con A-induced IL-6, a cytokine produced also by the cells of innate immunity [20], was less pronounced than that of IFN-γ (Figure 2A,B). S-in protein also significantly reduced proliferation of both CD4 and CD8 T cells, as indicated by CFSE assay (Figure 2C), supporting the restriction of polyclonal activation of the T cell compartment. Photographs of cell cultures at 72 h confirmed the lower activation of PBMC when treated with Con A in the presence of S-in, as indicated by the reduced number and size of Con A-induced agglomerates and clusters of activation (Figure 2D). These inhibitory effects were not due to toxicity, as shown by the lack of cell death in Figure 2E, and above all, by the cell viability XTT assay showing that Con A-stimulated cells, cultured in the presence or absence of the S-in protein, had comparable cellular mitochondrial metabolic activity (Figure 2E). Overall, the data suggested that the S-in protein, through its binding to Con A molecules, prevented the full ability of lectin to activate lymphocytes without affecting the PBMC metabolic activity. Collectively, the data of this and the previous section strongly suggest that the inhibition of Con A-induced, PBMC immune-activation and cytokine production is mediated by the glycan constituents of the S-in protein.

### 3.3. S-hu Protein and Its Fractions S1 and S2, as Well as S. cerevisiae Mannan, Increase IFN-γ and IL-6 Production by Con A-Stimulated PBMCs

We subsequently examined whether the results described in the previous section could be replicated with the S-hu, a protein obtained as a recombinant in cells of human origin (HEK293 cells). Surprisingly, the S-hu protein enhanced rather than inhibited cytokine production in Con A-stimulated PBMC, in apparent contrast with that observed with the S-in protein at equal doses of the two proteins. As shown in Figure 3, the S-hu protein, as well as its monomeric fractions S1 and S2, increased, in the Con A-stimulated PBMC of all donors assayed, the production of both IFN-γ and IL-6 (Figure 3A,B) and produced morphological changes of cultured cells consistent with increased cell proliferation/activation (Figure 3C). The S-hu slightly increased the mitochondrial metabolic activity of cells when compared to those cultured in medium alone, while Con A-stimulated cells, cultured in the presence or absence of the S-hu protein, had a comparable metabolic activity, as measured by a cell viability XTT assay (Figure 3D). In these cultures, the yeast mannan strongly promoted the production of both cytokines as well as cell activation (Figure 3C,E).

### 3.4. Both S Proteins, as Well as Yeast Mannan, Increase Cytokine Production by T Cells Activated with an Agonist Anti-CD3 Antibody

We then asked whether S proteins and mannan could also modulate non-lectin-induced polyclonal T cell responses. Thus, PBMC were stimulated with a low dose (0.8 ug/mL) of an agonist anti-CD3 monoclonal antibody (mAb). This resulted in production of IFN-γ, and, to a lower extent, IL-6, with important donor-to-donor differences (range 32–10,918 pg/mL for IFN-γ and 68–1874 pg/mL for IL-6). Of note, soluble anti-CD3 mAb without agonist anti-CD28 mAb, as in our conditions, required signals from monocytes for full T cell activation [21]. Both S proteins, or the S1 and S2 subunits, as well as mannan, increased both IFN-γ and IL-6 production and cellular activation induced by the anti-CD3 mAb (Figure 4A–D). Mannan showed the strongest co-stimulatory effect (Figure 4D), and S-in had a greater effect than S-hu protein and its subunits in the increase in anti-CD3-induced IL-6 (Figure 4B).

To evaluate whether the S protein and mannan also modified memory antigenic T cell responses, the PBMC were stimulated with the recombinant non-glycosylated *Candida albicans* MP65 protein, an immunodominant antigen used to ascertain the state of memory antigenic responses in human populations [22]. As expected, the antigen stimulated cytokine production in all donors tested, although with variability in response magnitude among donors (Figure 5A,B). In any case, regardless of the lesser or greater response to MP-65, neither S proteins nor mannan affected to any significant extent cytokine production (Figure 5A,B).

### 3.5. Both S-in and S-hu Proteins, as Well as Mannan, Induce IL-6, but Not IFN-γ Production by Unstimulated PBMCs

No antigen-specific immune responses are expected in cultures of the S protein–treated PBMC of pre-COVID-19 subjects, as were our donors. Nonetheless, our data showed a degree of metabolic activation of resting PBMC in the presence of S protein, as well as the capacity of S proteins to co-stimulate primary polyclonally activated T cell responses. This incited us to ask whether the S protein had itself some ability to induce cytokine production. Mannan was also used in these experiments. At 96 h of cell culture, no IFN-γ production was detected in unstimulated cells under medium conditions, nor in cells stimulated with S proteins or mannan, all values were around to 0 pg/ml. Instead, although not in all donors, both S-hu and, more markedly, S-in proteins, induced some IL-6 production at 96 h of cell culture (Figure 6A). The monomeric S1 and S2 subunits showed a very low effect (Figure 6A). Interestingly, under these culture conditions, mannan was the most potent inducer of IL-6 (Figure 6B,C) and, in combination with both S proteins, IL-6 production was further increased (Figure 6C). This suggested that IL-6 accumulation was mediated not exclusively by molecular interactions with mannan moieties. Since IL-6 is produced also by innate cells [20], a potential involvement of non-T cells, perhaps the monocytic cells, through their carbohydrate receptors, can be supposed. Mannan, S-in protein and, to a lower extent, S-hu protein caused some morphological changes (numbers and size of cell agglomerates) in PBMC cultures after 96 h of culture, consistent with some cellular activation if compared to medium-only cultures (Figure 6D).

### 3.6. Both S-in and S-hu Proteins Upregulate Mannose Receptor (CD206) Expression on Macrophages

The data of the previous sections suggested that the S protein of SARS-CoV-2 could interact with lectin-binding receptors present in non-T cells. Among these receptors, the CD206 (mannose receptor, MR), expressed predominantly by innate immune cells of monocytic and dendritic lineage, plays an important role in phagocytosis, endocytosis, immune homeostasis and protection against microbial pathogens [23]. CD206 is a prototypical marker of M2-type macrophage activation, characterized by IL-6 production among other cytokines [24]. Therefore, we investigated whether the S proteins modulated the expression of CD206 in the cultures of PBMC donors. Although not in all donors, we found that both S proteins, incubated with PBMC for 72 h, greatly upregulate CD206 expression on CD14^+^ monocytes/macrophages, as shown by flow cytometry analysis (Figure 7A,B). Stimulation of PBMC cultures with mannan for 72 h resulted in a similar effect (Figure 7A,B). Combination of mannan with S-in or S-hu protein did not modify the mannan-induced upregulation of CD206 after 72 h of culture (Figure 7B middle and right plots), suggesting that the S protein-mediated MR upregulation was likely dependent on their glycan moieties and on the interaction with MR itself. To evaluate the upregulation of the CD206 receptor in a context where the de novo protein synthesis of the receptor had a limited role, as well as the release of possible mediators by PBMCs cultured in the presence of S proteins or mannan, we added the stimuli in the last 30 min of the 72 h of culture. In this context, CD206 upregulation was more strongly induced by the S-in protein, suggesting its greater ability to interact with MR than that of the S-hu protein (Figure 7C and Appendix A).

## 4. Discussion

In this paper, we report findings that highlight the immunomodulatory capacity of the glycan ectodomain of SARS-CoV-2 S proteins generated in insect or human embryo cells, in cultures of human PBMC from pre-pandemic subjects: (*i*) both S proteins increase IL-6 and IFN-γ production in anti-CD3 mAb-stimulated T cells; (*ii*) the cell source (insect or human) of S protein determines up-and down-modulation of T cell activation and cytokine production induced by the mitogenic lectin Con A; (*iii*) the S proteins do not significantly impact cytokine production in memory antigenic responses triggered by a common microbial antigen; (*iv*) S proteins and the polysaccharide mannan (a polymer, consisting of alpha 1,6 linked mannose chains with alpha 1,2 mannose branches, obtained from the yeast *S. cerevisiae*) induce a non-neglectable IL-6 production, probably of non-T cell source, in the unstimulated PBMC cultures of some donors; (*v*) both the human and, more prominently, the insect S protein cause remarkable expression of CD206, a mannose receptor, which is expressed on typical cells of innate immunity such as the macrophages, dendritic and endothelial cells [25].

Although we have no direct evidence of a role of the glycan constituents present in the S protein, the so-called S protein ectodomain, in the aforesaid immunomodulatory activities, such a role is reasonably attributable to S glycan. In fact, we show here that the S protein strongly binds to Con A and this binding is inhibited by known Con A-ligands, such as the polysaccharide mannan and the alfa-methyl-mannoside. Interestingly, the polysaccharide mannan, substantially mimics, in our model, the immunomodulatory activities shown by the S proteins. It enhances cytokine production by anti-CD3-stimulated T cells, modulates Con A-induced responses, activates PBMC for the production of IL-6 (but not IFN-γ), and causes overexpression of the MR on CD14^+^ monocytes/macrophages. That the N-glycans of the S proteins are responsible for the immunomodulatory activity here reported is also suggested by the low activity of the S1 and S2 monomeric subunits, which have a lower portion of N-glycans when tested in comparison with the trimeric integral S protein at the same dosage. Moreover, the combination of mannan and S proteins does not result in an additive effect on upregulation of MR expression, suggesting that S proteins use their mannan moieties for MR engagement. Other data of ours are also supportive of a main role of the N-glycan molecules on S protein-induced immune-modulation in human PBMC cultures. We noticed a different potency of S-in than S-hu proteins at equal doses. Data from a number of authors have established that the N-glycan chains of the S-in protein are remarkably simpler and overall different, as expected, from those of the S-hu protein [26,27]. They are less complex and with low or no sialylation [26,27]. Differences in degree of polymerization and structure of the mannoside chains could be particularly relevant in explaining our differential data. To resolve this relevant question, we intend to assay in our models the S protein produced in *E. coli*, and therefore not glycosylated, and S-in or S-hu proteins with glycans modified by treatment with appropriate enzymes. In this context, the possible participation of N-glycan chains, particularly oligomannosides, spontaneously released from the S-protein backbone in PBMC cultures, cannot be ruled out.

Con A, a mannose/glucose-binding lectin isolated from Jack beans (*Concanavalia ensiformis*), is a well-known T cell mitogen that can activate the immune system, recruit lymphocytes and elicit cytokine production [18,19]. Specifically, binding of Con A triggers cross-linking of the TCR complex, leading to T cell activation, by activating NFAT (nuclear factor of activated T cells), a family of transcription factors relevant in the development and function of the immune system [28]. The glycan-binding capacity of lectins determines their mitogenicity, which relies on the lectin affinity for carbohydrates located on immune cell receptors. The Con A ability to activate T cells is also mediated by the activation of various co-stimulatory molecules, e.g., CD28, HEVM but also regulatory molecules such as CTL4 which, interacting with ligands expressed on antigen-presenting cells (e.g., CD80, CD86, CD160), provide or reduce secondary signaling [18,19,29]. The dosage of Con A and its interaction with co-stimulatory or regulatory molecules can determine both activation of effector function or trigger tolerance [30]. The different up- and down-modulation of Con A response induced by S-hu and S-in proteins could be due, in addition to a different affinity/avidity for Con A, to the differential activation of co-stimulatory or regulatory molecules concurring with T cell activation. Moreover, Con A, interacting with toll-like receptors 2 and 4, determines activation of the innate response, which could play a role in the production of IL-6 [31,32,33].

The activation of T lymphocytes with soluble anti-CD3 antibody induces cross-linking with the TCR, but also in this case the activation of a primary response is supported by second signals mediated by co-stimulatory molecules, leading to formation of the immunological synapse between T cells and antigen-presenting cells [21,34]. The glycoproteins carrying N- and O-linked oligosaccharides play a role in controlling the assembly and stabilization of the complexes in the immunological synapse as well as in protecting them from proteolysis during prolonged T-cell engagement [34]. Therefore, it is reasonable to assume that the S proteins and mannan itself enhance the anti-CD3 mAb response by favoring co-stimulation and the formation of the immunological synapse. However, further studies are required to identify the receptors/ligands expressed on T cells or other immune cells involved in the S protein-mediated modulatory effects on mitogenic activation of T cells by Con A or soluble anti-CD3 mAb.

The inability of S proteins to modulate the antigenic T cell memory response such as those elicited against a common antigen such as the MP65 of *C. albicans*, a known human commensal [22,35], may be due to the decreased need for co-stimulation and second signals in the reactivation of an antigenic memory response, as compared to the induction of a primary T response [36,37].

Overall, our data on the overproduction of proinflammatory cytokines mediated by the S proteins deserve particular consideration in patients treated with anti-CD3 monoclonal antibodies [38] and in all conditions in which saccharide-mediated co-stimulatory signals are relevant for T cell activation after TCR commitment. Similar caution should be used for experimental Con A-based anticancer therapies [33,39] in patients immunized with the current S-protein-based SARS-CoV-2 vaccines or affected by COVID-19. In addition, potential differences between COVID-19 protein vaccine produced in insect cells [40], and COVID-19 vaccines based on RNA technologies, more similar to protein produced in HEK293, warrant study.

To our knowledge, our report is the first to show that the S protein of SARS-CoV-2 is able to enhance the CD206, MR expression, an identifying marker of anti-inflammatory M2 macrophages [41]. MR is a member of the C-type lectins that serves as a homeostatic receptor by binding and scavenging unwanted mannose N-linked glycoproteins or hormones from circulation [42]. It recycles continuously between the plasma membrane and endosomal compartments in a clathrin-dependent manner [43]. The MR is expressed at low levels during inflammation and at high levels during the resolution of inflammation, to ensure inflammatory agents are removed from the circulation only at the appropriate time [42]. In the context of SARS-CoV-2 infection, the MR could be relevant for the intracellular access of the virus to macrophages and endothelial cells and their immune activation, suspected to be critical in particularly aggressive forms of COVID-19. Many pathogenic microbes are coated with mannose-containing structures, and the macrophage MR contributes to the intracellular access and phagocytosis of microbial pathogens [25,44]. Finally, the extracellular domain of MR is easily solubilized by the metallo proteinases, and this soluble MR induces metainflammation, even enabling IL-6 production by macrophages [45]. Of note, S-in protein shows a greater ability to interact with the MR compared to S-hu protein, similarly to what is observed in the binding between S proteins with circulating MLB [14]. In addition, S-in protein is also more powerful in inducing IL-6 by unstimulated PBMC. Further studies are warranted to determine the possible roles of MR and its soluble form in SARS-CoV-2 infection and COVID-19 disease.

We recognize that our study has a number of limitations. First, a relatively low number of human PBMC donors has been employed, and, as expected, there were non-neglectable, donor-to-donor differences in the studied immune responses. This was somewhat tempered by a clear, statistically assessable trend in the donor populations, not blurred by the inevitable background donor differences. Second, no direct proof of the S-glycan constituent involvement in the observed immunomodulatory effects is provided here. Nonetheless, the whole set of our observations clearly suggests it has a major role in most if not all the reported effects. Third, only two pro-inflammatory cytokines have been studied, and consequently the immunomodulatory setting we describe here, though focused upon two relevant cytokines in COVID-19 pathology, remains partial.

In conclusion, our data report novel findings that extend our knowledge of the S protein immunobiological activities, highlight the immunological mediation of its glycan ectodomain, and invite further studies to determine their significance in the fight against SARS-CoV-2 and the diseases it causes.

## Figures and Tables

**Figure 1 viruses-16-00497-f001:**
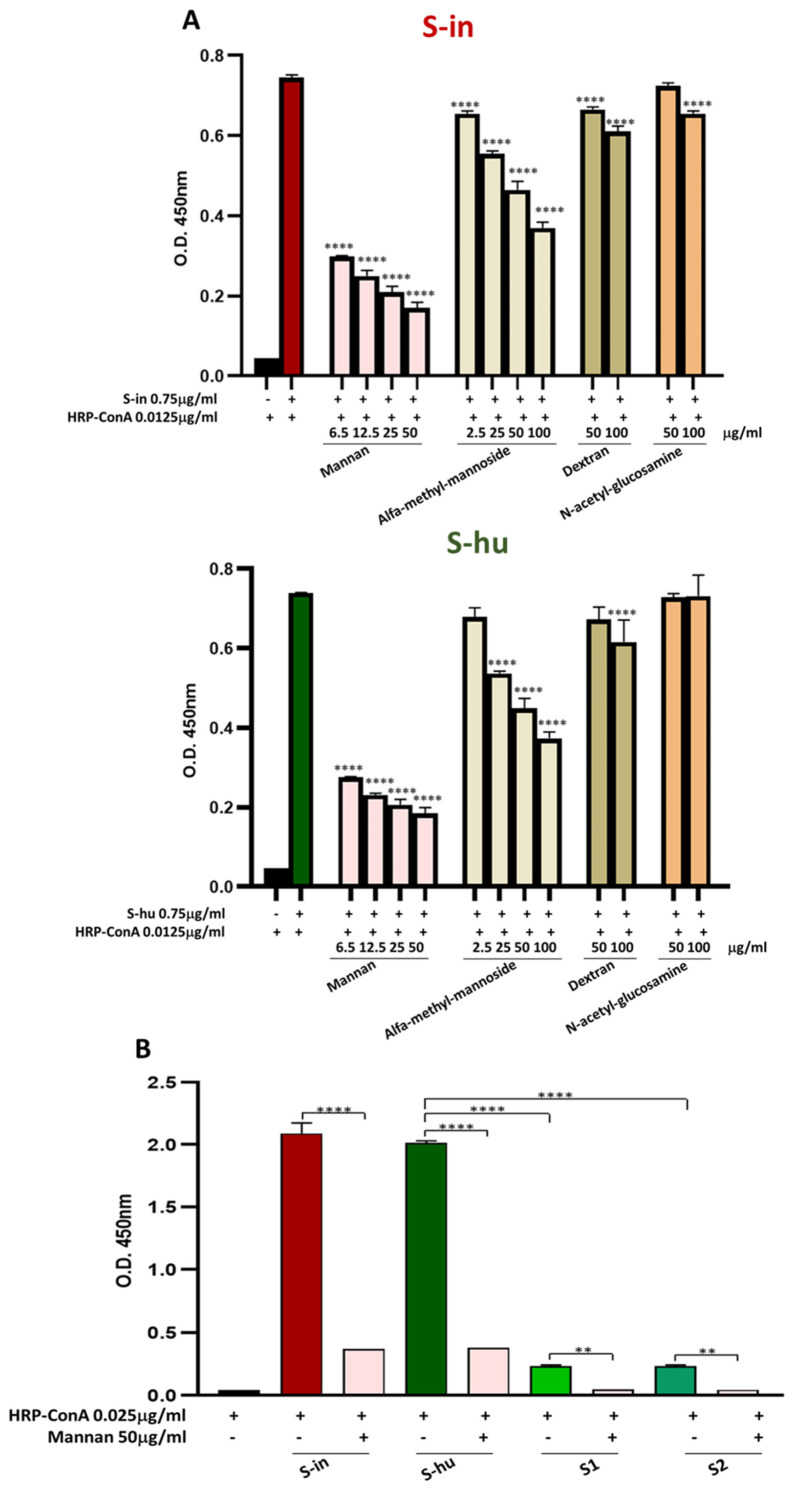
Spike (S) proteins bind Con A mainly through mannan constituents. (**A**) Binding of S-in (top) or S-hu (bottom) protein (0.75 µg/mL) coated on a microplate and HRP-Con A (0.0125 µg/mL), measured by optical density (O.D.) using an in-house ELISA assay. Scalar concentrations of saccharides were incubated together with HRP-Con A directly on the S protein-coated microplates. Data are reported as mean ± sd of two independent experiments run in duplicate. (**B**) Binding of S-in, S-hu, S1 and S2 proteins (all at 0.75 µg/mL) coated on a microplate and HRP-Con A (0.025 µg/mL) measured by optical density (O.D.) through an in-house ELISA assay. Mannan (50 µg/mL) was incubated together with HRP-Con A directly on the spike-coated microplates. Data are reported as mean ± sd of two independent experiments run in duplicate. Statistical analysis: one-way ANOVA with Tukey’s multiple comparisons post hoc test. ** *p* < 0.01; **** *p* < 0.0001. In panel (**A**), the asterisks indicate the differences with respect to the control condition Con A without saccharides. In panel (**B**), only differences in O.D. values between binding of Con A to each specific S protein in the absence or presence of mannan, and between binding of Con A and S-hu protein versus those of Con A and S1 or S2 subunits, are shown.

**Figure 2 viruses-16-00497-f002:**
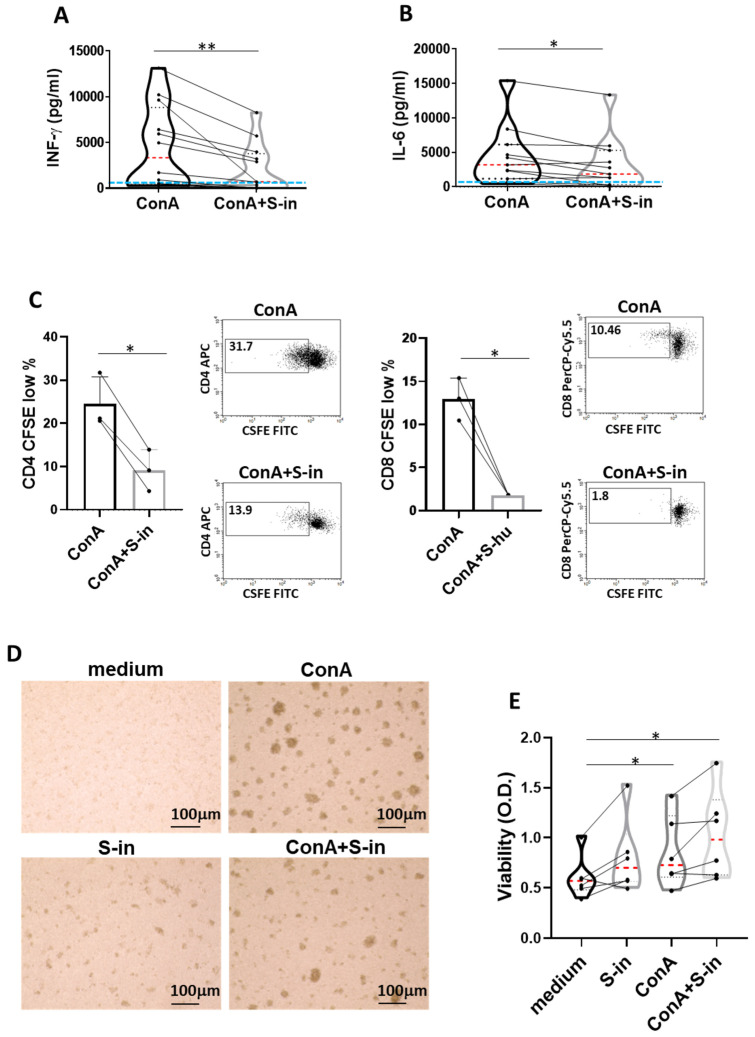
S-in protein prevents Con A-mediated activation of PBMCs recovered from blood of healthy donors in pre-SARS-CoV-2 pandemic times. PBMCs recovered from blood of healthy donors in pre-SARS-CoV-2 pandemic times were stimulated or not with Con A (4 µg/mL) in the presence or absence of S-in protein (5 µg/mL) before assaying cell proliferation at 72 h and cytokine production and cell viability at 96 h of culture. To measure T cell proliferation, PBMC were stained with CFSE before being cultured. (**A**,**B**) Violin plots combined with individual data plot (twelve donors for panel (**A**) and eleven for panel (**B**)) show protein levels of IFN-γ (**A**) and IL-6 (**B**) in culture supernatants of PBMC. The light blue line indicates the level of IFN-γ or IL-6 found in unstimulated PBMC cultured in medium only. Cytokines were quantified by specific ELISA kits and tested in duplicate. (**C**) Percentages of CFSE^low^ cells, as measures of CD4^+^ and CD8^+^ T cell proliferation. Representative density plots gated on CD3^+^CD4^+^ or CD3^+^CD8^+^ T lymphocytes are shown. Data were analyzed by Cell Quest software. Histograms are mean ± sd of three donors. Individual data points of each donor are also shown. (**D**) Representative images of PBMC cultures in the presence or absence of various stimuli and their combinations. (Magnification ×10, scale: 100 µm.) (**E**) Live metabolic active PBMC were measured through a colorimetric XTT assay. The violin plots combined with individual data plots indicated the OD value of formazan dye formed by the activity of mitochondrial dehydrogenases in the sample. Six individual donors were assayed in duplicate. In the violin plot, the red line indicates the median value while the two black dotted lines represent the first and third quartiles. The lines connecting data points in individual data plots are from the same donor. Statistically significant differences between groups were determined by Wilcoxon matched-pairs test, in panels (**A**,**B**,**E**) (in panel (**E**), the analyses were performed comparing the various groups two by two) and by paired *t* test in panel **C**. * *p* < 0.05; ** *p* < 0.01.

**Figure 3 viruses-16-00497-f003:**
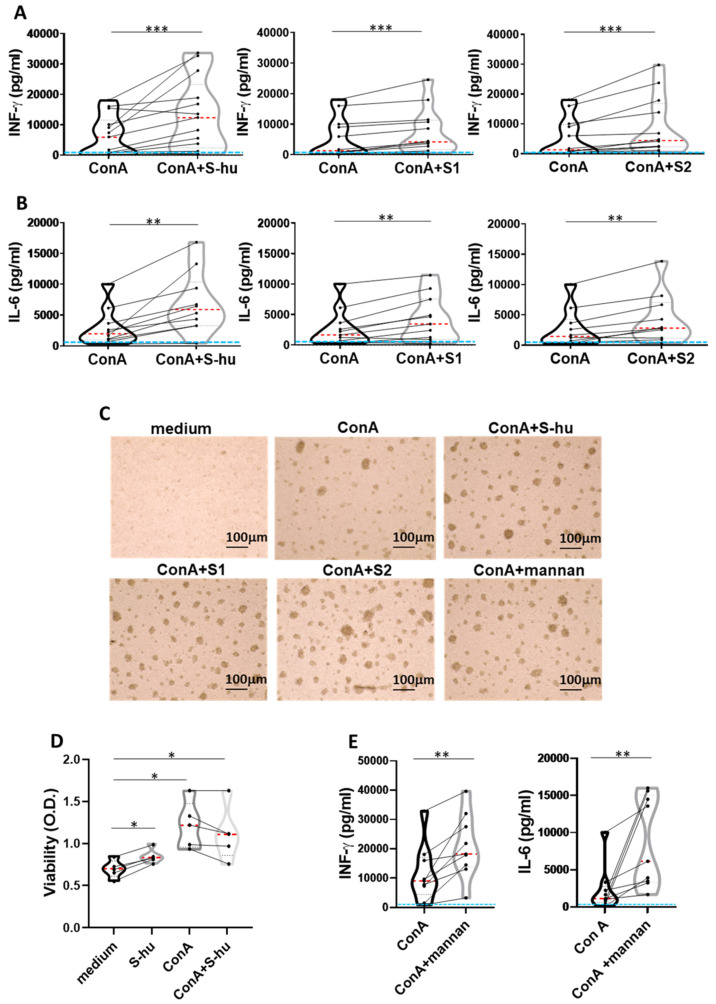
S-hu protein and its S1 and S2 subunits, as well as *S. cerevisiae* mannan, increase IFN-γ and IL-6 production by Con A-stimulated PBMCs. PBMCs recovered from blood of healthy donors in pre-SARS-CoV-2 pandemic times were cultured or not with Con A (4 µg/mL) in the presence or absence of S-hu, or S1 or S2 proteins (5 µg/mL), or mannan from *S. cerevisiae* (10 µg/mL) for 96 h. (**A**,**B**) Violin plots combined with individual values, showing protein levels of IFN-γ (**A**) and IL-6 (**B**) in culture supernatants of PBMC stimulated with Con A in the presence or absence of S proteins. Thirteen donors for S-hu protein, and twelve donors for both S1 and S2 in panel (**A**); eleven donors for S-hu protein, and ten donors for both S1 and S2 subunits, in panel (**B**) were assayed in duplicate. (**C**) Representative images of PBMC cultures in the presence or absence of various stimuli and their combinations (Magnification ×10, scale: 100 µm). (**D**) Live metabolic active PBMC were measured through a colorimetric XTT assay. The violin plots combined with individual data plots indicated the OD value of formazan dye formed by the activity of mitochondrial dehydrogenases in the sample. Five individual donors were assayed in duplicate. (**E**) Violin plots combined with individual values, showing protein levels of IFN-γ and IL-6 in culture supernatants of PBMC stimulated with Con A in the presence or absence of mannan. Nine donors were assayed in duplicate. All cytokines were quantified by specific ELISA kits and tested in duplicate. In the violin plots, the red line indicates the median value while the two black dotted lines represent the first and third quartiles. The lines connecting data points in individual data plots are from the same donor. The light blue line indicates the level of IFN-γ or IL-6 found in unstimulated PBMC cultures. Statistically significant differences between groups in panels (**A**,**B**,**D**) were determined by Wilcoxon matched-pairs test. * *p* < 0.05; ** *p* < 0.01; *** *p* < 0.001.

**Figure 4 viruses-16-00497-f004:**
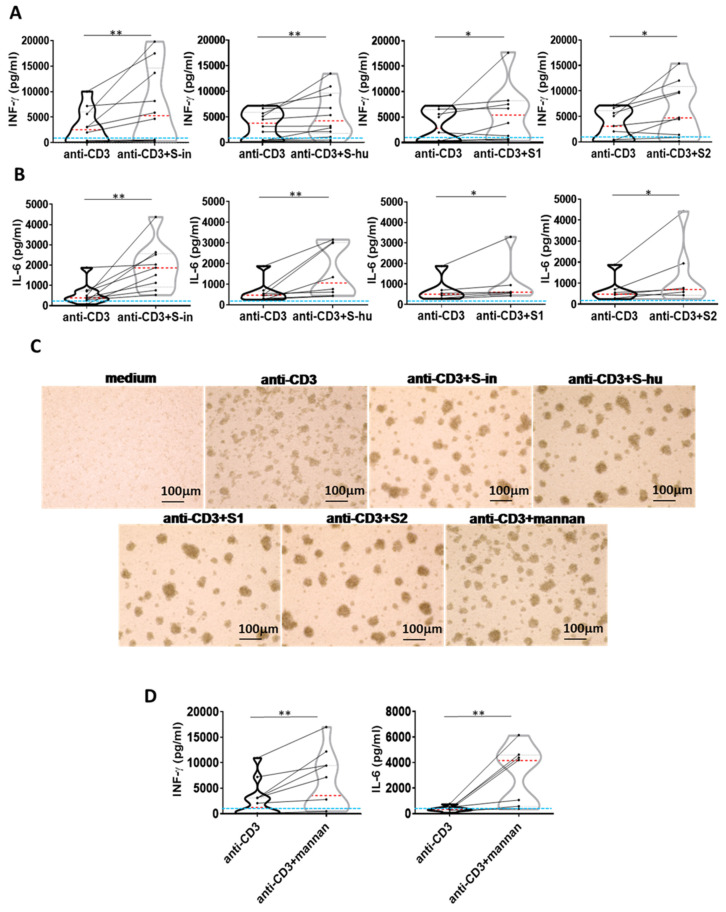
All S proteins, as well as *S. cerevisiae* mannan, increase IFN-γ and IL-6 production by anti-CD3 Ab-stimulated PBMCs. PBMCs recovered from blood of healthy donors in pre-SARS-CoV-2 pandemic times were cultured or not with an agonist anti-CD3 Ab (0.8 µg/mL) in the presence or absence of S-in, S-hu, S1 or S2 proteins (all at 5 µg/mL), *S. cerevisiae* mannan (10 µg/mL), or mannan and S protein combination for 96 h. (**A**,**B**) Violin plots combined with individual values, showing protein levels of IFN-γ (**A**) and IL-6 (**B**) in culture supernatants of PBMC stimulated with anti-CD3 Ab in the presence or absence of spike proteins. Ten donors for S-in protein, twelve for S-hu protein, nine donors for S1 subunit and nine donors for S2 subunit in panel (**A**); nine donors for S-in protein, eight donors for S-hu protein, six donors for S1 subunit and seven donors for S2 subunit, in panel (**B**), were assayed in duplicate. (**C**) Representative images of PBMC cultures in the presence or absence of various stimuli and their combinations. (Magnification ×10, scale: 100 µm.) (**D**) Violin plots combined with individual values, showing protein levels of IFN-γ and IL-6 in culture supernatants of PBMC stimulated with anti-CD3 Ab in the presence or absence of mannan. Seven donors for IFN-γ and IL-6 were assayed in duplicate. All cytokines were quantified by specific ELISA kits and tested in duplicate. In the violin plots, the red line indicates the median value while the two black dotted lines represent the first and third quartiles. The lines connecting data points in individual data plots are from the same donor. The light blue line indicates the level of IFN-γ or IL-6 found in unstimulated PBMC cultures. Statistically significant differences between groups were determined by Wilcoxon matched-pairs test; * *p* < 0.05; ** *p* < 0.01.

**Figure 5 viruses-16-00497-f005:**
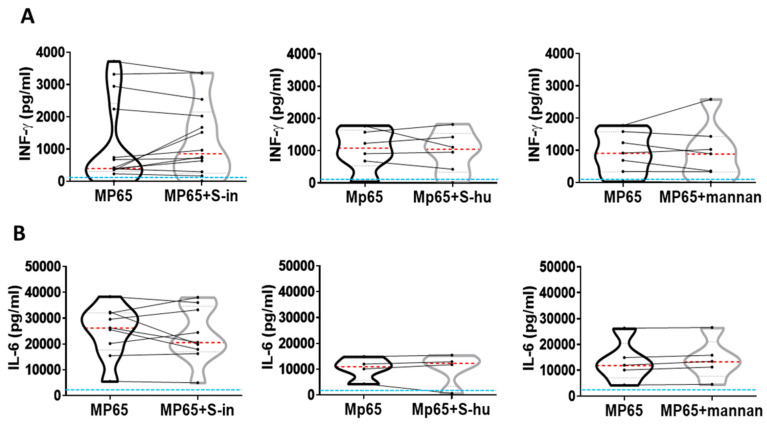
Neither S proteins nor mannan affect cytokine production by *C. albicans* antigen MP65-stimulated PBMCs. PBMCs recovered from blood of healthy donors in pre-SARS-CoV-2 pandemic times were cultured or not with recombinant non-glycosylated *Candida albicans* MP65 protein (1 µg/mL) in the presence or absence of S-in or S-hu proteins (all at 5 µg/mL) or *S. cerevisiae* mannan (10 µg/mL) for 96 h. Violin plots combined with individual values show protein levels of IFN-γ (**A**) and IL-6 (**B**) in culture supernatants. Fourteen donors for S-in protein, six donors for S-hu protein and seven donors for mannan in panel (**A**); ten donors for S-in protein, four donors for S-hu protein and five donors for mannan in panel (**B**) were assayed in duplicate. Cytokines were quantified by specific ELISA kits and tested in duplicate. In the violin plot, the red line inside the violin plot indicates the median value, while the two black dotted lines represent the first and third quartiles. The lines connecting data points in individual data plots are from the same donor. The light blue line indicates the level of IFN-γ or IL-6 found in unstimulated PBMC cultures. Statistically significant differences between groups were determined by Wilcoxon matched-pairs test.

**Figure 6 viruses-16-00497-f006:**
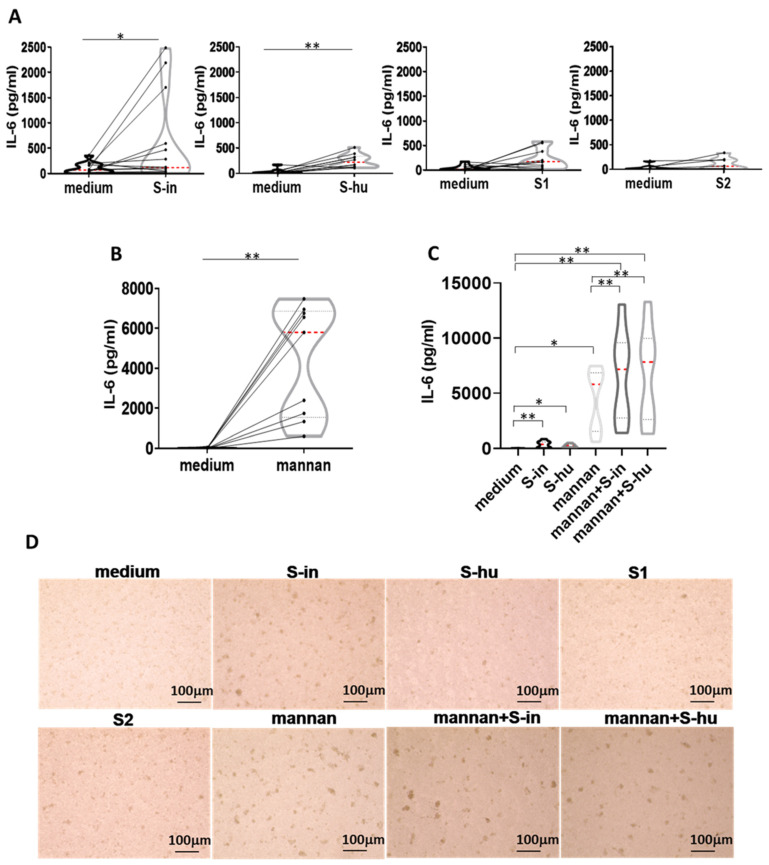
Both S-in and S-hu proteins, as well as mannan, induce IL-6 production by unstimulated PBMCs. PBMCs recovered from blood of healthy donors in pre-SARS-CoV-2 pandemic times were cultured in the presence or absence of S-in, S-hu, S1 or S2 proteins (all at 5 µg/mL) or S. cerevisiae mannan (10 µg/mL) or combination of S proteins and mannan for 96 h. (**A**) Violin plots combined with individual values, showing protein levels of IL-6 in culture supernatants of medium-cultured PBMC in the presence or absence of spike proteins. Thirteen donors for S-in protein, ten donors for S-hu protein, eleven donors for S1 subunit and seven donors for S2 subunit were assayed in duplicate. (**B**) Violin plots combined with individual values, showing protein levels of IL-6 in culture supernatants of medium-cultured PBMC in the presence or absence of mannan. Nine donors were assayed in duplicate. (**C**) Violin plots combined with individual values, showing protein levels of IL-6 in culture supernatants of medium-cultured PBMC in the presence or absence of S-in protein, S-hu protein or mannan and their combinations. Eight donors were assayed in duplicate. All cytokines were quantified by specific ELISA kits and tested in duplicate. The red line inside the violin plots indicates the median value, while the two black dotted lines represent the first and third quartiles. The lines connecting data points in individual data plots are from the same donor. (**D**) Representative images of PBMC cultures in the presence or absence of various stimuli and their combinations. (Magnification ×10, scale: 100 µm.) Statistically significant differences between groups were determined by Wilcoxon matched-pairs test. In panel (**C**), the analysis was performed comparing the various groups two by two; * *p* < 0.05; ** *p* < 0.01.

**Figure 7 viruses-16-00497-f007:**
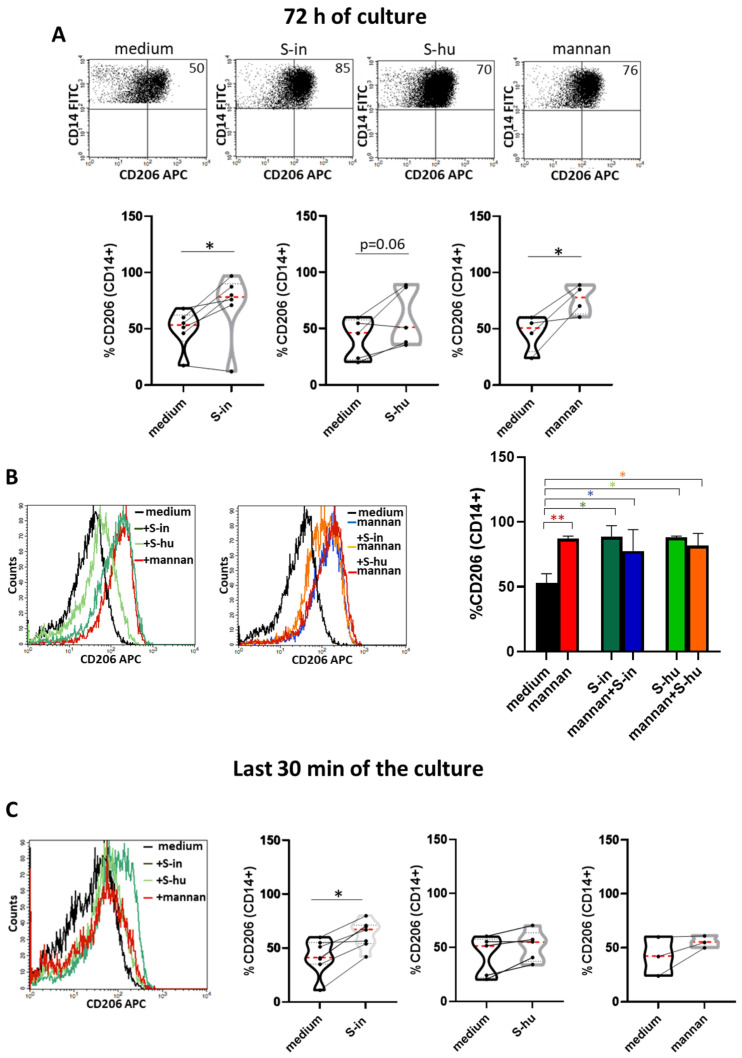
S proteins as well as mannan upregulate mannose receptor expression on CD14+ monocytes/macrophages. PBMCs recovered from blood of healthy donors in pre-SARS-CoV-2 pandemic times were cultured in the presence or absence of S-in and S-hu proteins (all at 5 µg/mL), *S. cerevisiae* mannan (10 µg/mL) or their combination for 72 h before measuring the expression of CD206 on CD14^+^ monocytes/macrophages by flow cytometry. In some conditions, stimuli were added for the last 30 min of culture. (**A**) Representative dot plots for CD206 expression on CD14+ cells of PBMC cultured in the presence or absence of S-in, S-hu or mannan for 72 h. Violin plots combined with individual values showing the CD206 expression on CD14^+^ cells of different donors. Six donors for S-in, five donors for S-hu and four donors for mannan were assayed. (**B**) Representative histogram plots for CD206 expression on CD14^+^ of PBMC cultured for 72 h with S-in protein, S-hu protein and mannan (left plot) or their combination (middle plot). The bars in the histogram (right plot) represent the percentage of CD14+ cells positive for the CD206 marker as mean ± sd of 3 donors. (**C**) Representative histogram plot for CD206 expression on CD14^+^ of PBMC cultured for 72 h and stimulated with S-in protein, S-hu protein or mannan for the last 30 min of culture (left panel). Violin plots combined with individual values showing the CD206 expression on CD14^+^ of PBMC cultured in medium for 72 h and stimulated or not with S-in or S-hu proteins or mannan in the last 30 min of culture. Six donors for S-in, five donors for S-hu and four donors for mannan were assayed. The red line inside the violin plots indicates the median value, while the two black dotted lines represent the first and third quartiles. The lines connecting data points in individual data plots are from the same donor. All data were analyzed by Cell Quest software. Statistically significant differences between groups were determined by Paired *t* test, in panel (**A**,**C**) and by one-way ANOVA with Tukey’s multiple comparisons post hoc test in panel (**B**). * *p* < 0.05; ** *p* < 0.01.

## Data Availability

The data that support the findings of this study are available from the corresponding authors upon reasonable request.

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
