# Peer review of "The Glycan Ectodomain of SARS-CoV-2 Spike Protein Modulates Cytokine Production and Expression of CD206 Mannose Receptor in PBMC Cultures of Pre-COVID-19 Healthy Subjects"

_viruses, 2024, doi:10.3390/v16040497_

Round 1

Reviewer 1 Report

Comments and Suggestions for Authors

Barbati and colleagues investigated the modulatory effect of the SARS-CoV-2 spike protein on the production of cytokines and the expression of the mannose receptor CD206 in PBMCs. There is evidence that SARS-CoV-2 might act as an inflammation inducer; further, the release of cytokines can lead to a severe outcome of COVID-19. Mechanisms involved in the inflammatory activity of SARS-CoV-2 S are so far poorly understood. Here, the authors investigated the ability of SARS-CoV-2 S generated in insect or human cell lines to bind to concanavalin A and to prevents Con A-induced PBMC activation. Further, the cytokine production of stimulated and unstimulated PBMCs was analysed in the presence and absence of SARS-CoV-2 S.

In general, the manuscript is well written and provides novel information with regard to SARS-CoV-2 induced PBMC activation and cytokine production. The results are presented in a clear fashion and include necessary controls.

The following points should be addressed prior acceptance of the manuscript:

1.       Line 90-96: Based on the catalogue numbers given in the methods, it was not possible to get detailed information on the spike proteins used for this study. E.g. products 40589-V08H 681AA and 40589-V08H1 511 are not listed by SinoBiological. For S-hu and S-in, it is not clear which version has been used for this study. Please check the product numbers given in the text.

2.       Please provide information on the isolate/sequence of S. Are both spike proteins derived from the same isolate? Are they codon optimized?

3.       Fig. 1A: Data for S-hu should be included.

4.       Fig. 3: Are data on viability/ mitochondrial metabolic activity available?

5.       Fig. 7A and C: Please include data for S-hu.

Author Response

REVIEWER 1

Comments and Suggestions for Authors

Barbati and colleagues investigated the modulatory effect of the SARS-CoV-2 spike protein on the production of cytokines and the expression of the mannose receptor CD206 in PBMCs. There is evidence that SARS-CoV-2 might act as an inflammation inducer; further, the release of cytokines can lead to a severe outcome of COVID-19. Mechanisms involved in the inflammatory activity of SARS-CoV-2 S are so far poorly understood. Here, the authors investigated the ability of SARS-CoV-2 S generated in insect or human cell lines to bind to concanavalin A and to prevents Con A-induced PBMC activation. Further, the cytokine production of stimulated and unstimulated PBMCs was analysed in the presence and absence of SARS-CoV-2 S.

In general, the manuscript is well written and provides novel information with regard to SARS-CoV-2 induced PBMC activation and cytokine production. The results are presented in a clear fashion and include necessary controls.

The following points should be addressed prior acceptance of the manuscript:

  1. Line 90-96: Based on the catalogue numbers given in the methods, it was not possible to get detailed information on the spike proteins used for this study. E.g. products 40589-V08H 681AA and 40589-V08H1 511 are not listed by SinoBiological. For S-hu and S-in, it is not clear which version has been used for this study. Please check the product numbers given in the text.

REPLY: All product numbers of the recombinant Spike proteins used in this work are correct and visible on the Sino biological website. For greater clarity, we have rewritten the entire name of each Sars-Cov-2 (2019-nCoV) Spike recombinant protein in full and moved the product number after product information. Lines: 90-101

  1. Please provide information on the isolate/sequence of S. Are both spike proteins derived from the same isolate? Are they codon optimized?

REPLY: The two Sars-Cov-2 (S1+S2) spike recombinant proteins as well as the S1 and S2 subunits used in this work derived from the same isolate (2019- nCoV) and were His-tagged. Lines 90-101.

  1. 1A: Data for S-hu should be included.

REPLY: We agree with the reviewer and we prepared a new Figure 1A with data including those of S-hu protein. We have also added a new supplementary figure (Supplementary Figure S1) to better compare the percentages of inhibition exerted by various mannan or alpha-methyl-mannoside concentrations on the S-in-Con A bond or S-hu-Con A bond. The text in the results section (Lines 204-212) and the legend of Figure 1 have been accordingly modified (Lines 220, 222)

  1. 3: Are data on viability/ mitochondrial metabolic activity available?

REPLY: We have shown in figure 3 the data on viability/mitochondrial activity (XTT assay) for S-hu (panel D). The text in the results section (Lines 294-297) and the legend of Figure 3 have been accordingly modified (Lines 310-312).

  1. 7A and C: Please include data for S-hu.

REPLY: We have shown in Figure 7A and C the FACS data for S-hu and mannan. Representative dot plots have also been added for all condition at 72h and last 30 min of culture. The last data have been shown in a new supplementary figure (Supplementary Figure S2). The text in the results section (Lines 431, 437-444) and the legend of Figure 7 have been accordingly modified (Lines 452-468).

Reviewer 2 Report

Comments and Suggestions for Authors

In this manuscript, the authors have discovered that the Sars-Cov-2 S protein can bind Con A through its glycan (mannosides) constituents. Additionally, they have identified that N-glycans have the ability to modulate the S protein's immunomodulatory effect via PBMC. The manuscript includes a variety of experiments such as In vitro toxicology assays, ELISA, and flow cytometry, with well-represented data supporting the conclusions drawn. However, there are several concerns that need to be addressed.

Major points:

1.Many published studies have reported that mannose can impact the function of T cells and macrophages. Therefore, it is crucial to design experiments that can eliminate the potential effects of undecorated mannose.

2.Concanavalin A is a mannose/glucose-binding lectin isolated from Jack beans. It is important to verify whether the Concanavalin A used in this study carried mannose.

3.Figure 7 indicates an increase in CD206 positive cells in the S-in and S-hu protein treated groups. Authors should consider performing q-PCR or Western blotting to determine the expression levels of CD206.

Minor points:

1.Authors should review the text within Figures 1, 2, 3, 4, and 5 for accuracy.

2.The images of PBMC may not be reader-friendly and could be improved for better understanding.

3.The results of Flow Cytometry presented in this study should adhere to guidelines.

Comments on the Quality of English Language

The English language still needs improvement.

Author Response

REVIEWER 2

Comments and Suggestions for Authors

In this manuscript, the authors have discovered that the Sars-Cov-2 S protein can bind Con A through its glycan (mannosides) constituents. Additionally, they have identified that N-glycans have the ability to modulate the S protein's immunomodulatory effect via PBMC. The manuscript includes a variety of experiments such as In vitro toxicology assays, ELISA, and flow cytometry, with well-represented data supporting the conclusions drawn. However, there are several concerns that need to be addressed.

Major points:

1.Many published studies have reported that mannose can impact the function of T cells and macrophages. Therefore, it is crucial to design experiments that can eliminate the potential effects of undecorated mannose.

 REPLY:  As we understand, this referee raises the possibility that the sugar mannose, if detached or naturally split from the oligomannoside chains which decorate the Spike protein, can induce, wholly or in part, the immunomodulatory effects we have observed and have attributed to the N-glycan of SARS-CoV-2 glycoprotein. While, to the best of our knowledge, mannose has no ability to activate human PBMC cells for IL-6 and IFN-g cytokine production, nonetheless we have given serious consideration to the above comment and decided to test directly the effect of mannose, at concentration (10 mg/ml), higher than that of S protein, on Con A-stimulated IFN-g production and on CD206 mannose receptor expression, as compared to S-protein stimulation, using PBMC of a donor included in our study.

As shown below, the addition of mannose did not cause a sizeable enhancement of the basic cytokine production of the Con A-stimulated PBMC culture whereas the addition of the S-human glycoprotein greatly increased the production confirming what we have shown in the paper

PBMC donor             Stimuli                                        IFN-g (pg/ml)

Donor 2                 ConA (4 mg/ml)                                7681 ±855

                              ConA + S-hu (5 mg/ml)                   17221±1250

                              Con A+ D-mannose (10 mg/ml)         8389±673

 Similarly, in an experiment of CD206 expression on CD14+ cells (using the same donor) the percentages of expression of this receptor (at 30 min) were   24, 39, 50 and 28 in cells treated with medium only, S-hu, mannan and mannose, respectively.

In conclusion, the data obtained by treating the PBMC cultures with mannose do not indicate that

this simple sugar is responsible of the effects we observed with the S glycoprotein. However, it cannot be definitely ruled out that oligomannoside chains, which especially decorate the S protein generated in insects, could participate in the immunomodulatory effects that we report here under conditions where, through enzymatic or other actions, are released from the S-glycoprotein molecule. We have added a sentence (Lines 509-511) specifying this in the revised version of the manuscript.

2.Concanavalin A is a mannose/glucose-binding lectin isolated from Jack beans. It is important to verify whether the Concanavalin A used in this study carried mannose.

REPLY: May we respectfully disagree with the reviewer on this comment. Since mannose is the hexose sugar that binds with the highest affinity to Con A, most if not all of the lectin – sugar binding sites would be occupied by mannose, then Con A would not be expected to bind efficiently the S protein. Instead, we show that Con A efficiently binds both S proteins in our paper.

3.Figure 7 indicates an increase in CD206 positive cells in the S-in and S-hu protein treated groups. Authors should consider performing q-PCR or Western blotting to determine the expression levels of CD206.

REPLY: We chose to evaluate the expression of C206 via flow cytometry for a series of reasons. First, and unlike other methods, cytofluorimetry has allowed us   to identify the CD206 expression on the CD14-positive monocyte/macrophage, a subset of the heterogeneous population of human PBMCs. Second, and above all, flow cytometry, differently from the other methods, allows to measure the expression of this receptor on the surface of cells, excluding the intracellular CD206. In fact, the mannose receptor recycles continuously between the plasma membrane and endosomal compartments in a clathrin-dependent manner (Gazi U, Martinez-Pomares L (2009). "Influence of the mannose receptor in host immune responses". Immunobiology. 214 (7): 554–61 doi:10.1016/j.imbio.2008.11.004).  Evaluating the surface expression of CD206 on CD14 positive cells was the aim of our analysis.

Minor points:

1.Authors should review the text within Figures 1, 2, 3, 4, and 5 for accuracy.

REPLY: We rephrased the figure legends

2.The images of PBMC may not be reader-friendly and could be improved for better understanding.

REPLY: We have tried to obtain more readable images in the revised version. Moreover, we explained the significance of morphological changes in the photos of PBMC cultures. See also Reply to reviewer 3 Results N° 16. Lines 249-251, 398-399

3.The results of Flow Cytometry presented in this study should adhere to guidelines.

REPLY: We now show representative dot plots for CD206 expression on CD14+ macrophages in Figure 7 and Supplementary Figure S2. In the Supplementary Figure we also show the gating strategy of our analysis.

Reviewer 3 Report

Comments and Suggestions for Authors

The spike protein of SARS-CoV-2 plays a critical role in viral infection and the progression of COVID-19, including severe forms such as cytokine release syndrome. However, potential direct effects of the spike protein on inflammation during disease progression is not fully understood. In this paper, Barbati and Bromuro et al. investigate the ability of glycosylated spike protein to stimulate pathways of inflammation and T cell activation pathways in pre-pandemic human donor PBMCs. Spike protein produced by human (S-hu) or insect (S-in) cells showed significant effects on IL-6 and IFN gamma production in the presence of Con A or anti-CD3 mAb. These effects largely mimicked stimulation when mannan was added, and further similarities in the upregulation of mannose receptor CD206 suggested that N-linked glycans on the spike surface may mediate these effects. Though limited in scope, this study offers interesting glimpses into the role of glycans in shaping immune responses to infection, particularly that glycan structure may be consequential.

Paper strengths:

1.       The overarching concept of this paper is quite interesting; namely, that spike, with much attention paid to its interactions with adaptive immunity, could have a concurrent and dramatic impact on inflammation. That sense of novelty should interest and would benefit readers.

2.       Overall, the research and conducted experiments are thorough and thoughtful. I appreciate the inclusion of spike produced from two different sources, which certainly increases the amount of data to collect, but adds greatly to the interest and soundness of the study.

Areas of improvement:

1.       Some of the figures show data presented in inconsistent ways without a clear reason, making some of the story in this manuscript feel incomplete. Representative images were intermittently included, sometimes for the same type of assay (e.g. flow cytometry). Several figures picture cells following stimulation, but cell viability is only explored for one figure. Notably, an entire analysis of Con A binding and interference is included for S-in (Figure 1A), but not for S-hu. Though I don’t expect adding data where suggested (see comments below) would dramatically change the story, it would improve transparency and ensure that readers see this research is carefully and completely shown. This may be especially important because the authors likely wanted to investigate S-hu and S-in equally, and did not seem to focus on one or the other because of certain criteria. I understand that this suggestion would add more pieces of data when the existing figures are already complex. However, the authors should strongly consider keeping the data presentation consistent and put less critical images in supplemental (e.g. representative images or whatever the authors choose).

2.       Some language in the description of results was a little vague, and should be clarified, either in legends or main text.

Introduction:

1.       Line 40: Good to change to severe acute respiratory syndrome 2 (SARS-CoV-2) as an initial definition for clarity

2.       Line 45: “unequally distributed and worldwide available” – best to cite and clarify here, maybe “available worldwide, but distributed unequally”?

3.       Lines 52-53: Could mention the receptor binding domain here – critical for binding to ACE2. Also need citation for function of ACE2.

4.       Lines 53-55: CRS citation needed – “ample evidence” but no attached references

5.       Line 61: First line, it can be clearer that you’re talking about N-linked glycans, potentially with some background on their synthesis? O-linked glycans could be mentioned, but maybe not required.

6.       Line 70-71: “that was” after the comma a little unclear – do you mean “which was”?

7.       Line 81: Was there a particular baculovirus cell line that was used for S-in production? Would be helpful to mention alongside hek293 at this stage.

Methods:

1.       Lines 91-93: Why would spike protein produced in baculovirus have an affinity (or ELISA binding? EC50 not quite clear on its own) of 400-1200, but spike protein from hek293 has a value ~40x higher (10-50)? Is ACE2 binding dependent on glycan structure?

2.       Line 109: ELISA had spike protein coated at 0.75 ug/ml – was the concentration taken from the literature, or did it happen to work best for your assay? Would help to briefly address here.

3.       Lines 134-137: Says “the following stimulus” and lists three different stimuli – are these separate stimulus experiments, or are all items needed for appropriate stimulus? A little more description here would help.

4.       Line 141: Speed and duration for extraction of supernatant?

5.       Lines 177-179: Reference manufacturer of cytometer and developer of software

Results:

1.       Line 191: Can focus on N-linked glycan moieties here, unless the references are talking about all kinds of glycans.

2.       Figure 1A: Binding to Con A and inhibition of Con A binding by several molecules is shown, but only for S-in. Was the same analysis conducted with S-hu? It should be included to initially demonstrate binding, as is shown with S-in.

3.       Figure 1A and 1B: The ug/ml units are a little hard to read in both figures – or does it say mcg/ml?

4.       Figure 1B: Surprised that S1 and S2 subunits bind to Con A at virtually the same level? Or maybe not, if the number of glycans on S1 and S2 is quite similar? Also, were these proteins produced in human or insect cells? Lastly, the S-hu, S1, and S2 portions of this panel have “+” for mannan and the protein of interest underneath every bar, which does not follow the pattern of +/- mannan in the S-in bars.

5.       Line 233: It says “Figure 1 A and B” as a reference, but the authors might mean Figure 2 A and B instead.

6.       Lines 239-240: In some cases, the OD from the XTT assay was comparable, but in some cases the differences are significant. Can you comment on why cell viability might be significantly different between these conditions?

7.       Figure 2A and 2B – overall the figures are good, but too small. These panels should be bigger to see all details clearly. Some elements of the panels are also unclear. It looks like there’s a dark blue line in each plot separate from the noted red and light blue lines. If kept, the significance of this line should be noted in the legend. Also, lines are connecting data points between groups, but it’s not clearly explained why. Are these lines indicating they are from the same donor? Any clarification in the legend would be helpful.

8.       Figure 2C and 2E – the previous figures represent individual measurements as dots within the plot, but the same is not done for replicate data (three for 2C, six for 2E?) in these figures. It would be helpful to include the same information here. Also, the lines in the violin plots of 2E are likely referring to the same metrics as 2A and 2B (red line = median, etc.), but it is not stated in the legend explicitly.

9.       Figure 3A, B, D – same comment about the dark blue/black lines in violin plot – unclear of its significance.

10.   Figure 3C – was the XTT assay used to measure cell viability for these images, and were any differences found?

11.   Line 304: The phrase “monocytes/macrophages” is a little unclear. Do you mean that both monocytes and macrophages are required, or that monocytes differentiating into macrophages are required?

12.   Lines 323-324/Figure 3D: The legend states that donors were assayed in duplicate, but that the data are plot from a run of “triplicate or quadruplicate”. More clarification on how data was obtained vs. how it is plotted would be helpful.

13.   Figure 5B: In the violin plot on the left, the median IL-6 protein level of donors with only MP65 stimulation is nearly three times higher than the median levels of MP65 alone in the other two panels. Was there anything different about the stimulation of these donors that would make the median so high compared to other groups? What range of measured IL-6 protein level might be expected in this assay following stimulation?

14.   Lines 358-360: If the data will not be shown for this point, it would be helpful to add more details about these particular IFN gamma experiments. For instance, “even late in the culture” is vague about the timepoint(s) where stimulation did not occur. I would encourage the authors to include violin plots for this data in the same fashion as other panels in this manuscript. Even showing results with no significant differences would be helpful for transparency, and could be put in supplemental if allowed by the journal.

15.   Line 365: The authors mention “late IL-6 accumulation,” but it’s not clear what late means in this context (at 96 hrs? before 96 hrs?). If applicable, referencing a specific timepoint in the assay would be more informative.

16.   Line 369: “Some morphological changes,” but no details are given by the authors. What morphological changes were observed, and how do they compare to previous panels under other stimulation conditions?

17.   Line 384: The red line showing the median values is mentioned, but the legend does not mention or describe the blue lines shown in violin plots of Figure 6B and 6C.

18.   Lines 409-411: Suggesting that S-in has a higher affinity or avidity to CD206, but do not show direct evidence of binding. Can this be tested directly with CD206? Also, it is unclear whether the authors suspect direct contact of CD206 with glycans on spike, boosting of expression through some indirect mechanism, or both.

19.   Figure 7A: A summary of CD206 positive cells with and without S-in is shown with violin plots, but a representative plot from the flow cytometer showing any gating, staining level, etc. (similar to Figure 2C) is not included. As with other pieces of data in this manuscript, it would help to show a representative image of flow CD206 positive % cytometry as well as the summary for transparency. Also, was this same experiment conducted with S-hu, and can those results be shown too?

20.   Figure 7B: The MFI of S-in appears to be a different level than the S-in MFI from Figure 7A. Is that the result of showing the mean instead of the median, or were different donors tested? A few more comments about this figure. First, the text of the axes for the panels showing counts and CD206 APC is quite small, and increasing the size would improve readability. Second, the histogram panel on the left does not indicate whether comparisons between the S-in, S-hu, and mannan groups are significant or not, as is shown in the panel on the right. Finally, the authors directly compare MFI of mannan with or without S-in or S-hu, but not the reverse. Can you test for significance there too (e.g. S-in in the left plot vs. mannan+S-in in the right plot)? If making those comparisons is possible, these panels could then be combined as well, making the figure a little less crowded.

21.   Figure 7C: Same comment as Figure 7A – a representative cytometry plot could help, even in supplemental

22.   Figure 7D: Tests for significance not shown among the S-in, S-hu, and mannan groups.

Discussion:

1.       Lines 463-465: “A number of authors” are mentioned to support this sentence, yet no citations to specific studies are given.

2.       Lines 466-467: The authors mention differences between glycan structures as a possible reason for differences in S-hu and S-in immunomodulation. Though exploring this hypothesis appears outside of the scope of this manuscript, this section would be more valuable if the authors discussed ways in which the effect of glycan structure could be directly investigated. For instance, have they thought about modifying glycans directly (e.g. stripping them off of glycosylated spike with an enzyme) or actively changing the composition of the glycans with GnTi- cells? Perhaps direct affinity measurements with CD206? I would encourage the authors to add their take on possible future directions.

3.       Lines 484-485: Reference on Con A interaction with toll like receptors would be helpful here.

4.       Line 491: “The sugars” – do you mean N-glycans specifically? If so, it would help to note here and include a citation for the sentence.

5.       Lines 507-512: The authors make an interesting point about potential interactions between vaccination with spike antigens and existing immunomodulatory therapies. Could this potential effect extend beyond COVID (thinking especially of immunization with influenza and RSV glycoproteins), or does spike protein have unique immunomodulatory effects that make this a concern? Commenting on this possibility could be informative and broaden the scope of suggested future directions.

6.       Lines 513-517: First two sentences of this paragraph introduce some points about CD206, but do not have citations.

Author Response

REVIEWER 3

Comments and Suggestions for Authors

The spike protein of SARS-CoV-2 plays a critical role in viral infection and the progression of COVID-19, including severe forms such as cytokine release syndrome. However, potential direct effects of the spike protein on inflammation during disease progression is not fully understood. In this paper, Barbati and Bromuro et al. investigate the ability of glycosylated spike protein to stimulate pathways of inflammation and T cell activation pathways in pre-pandemic human donor PBMCs. Spike protein produced by human (S-hu) or insect (S-in) cells showed significant effects on IL-6 and IFN gamma production in the presence of Con A or anti-CD3 mAb. These effects largely mimicked stimulation when mannan was added, and further similarities in the upregulation of mannose receptor CD206 suggested that N-linked glycans on the spike surface may mediate these effects. Though limited in scope, this study offers interesting glimpses into the role of glycans in shaping immune responses to infection, particularly that glycan structure may be consequential.

Paper strengths:

  1. The overarching concept of this paper is quite interesting; namely, that spike, with much attention paid to its interactions with adaptive immunity, could have a concurrent and dramatic impact on inflammation. That sense of novelty should interest and would benefit readers.
  2. Overall, the research and conducted experiments are thorough and thoughtful. I appreciate the inclusion of spike produced from two different sources, which certainly increases the amount of data to collect, but adds greatly to the interest and soundness of the study.

REPLY. We thank the reviewer for the general appreciation of our scientific effort.

Areas of improvement:

  1. Some of the figures show data presented in inconsistent ways without a clear reason, making some of the story in this manuscript feel incomplete. Representative images were intermittently included, sometimes for the same type of assay (e.g. flow cytometry). Several figures picture cells following stimulation, but cell viability is only explored for one figure. Notably, an entire analysis of Con A binding and interference is included for S-in (Figure 1A), but not for S-hu. Though I don’t expect adding data where suggested (see comments below) would dramatically change the story, it would improve transparency and ensure that readers see this research is carefully and completely shown. This may be especially important because the authors likely wanted to investigate S-hu and S-in equally, and did not seem to focus on one or the other because of certain criteria. I understand that this suggestion would add more pieces of data when the existing figures are already complex. However, the authors should strongly consider keeping the data presentation consistent and put less critical images in supplemental (e.g. representative images or whatever the authors choose).

REPLY: Following the reviewer's suggestion, the new data for the missing parts has been inserted in the new panels of Figures 1, 3 and 7. Furthermore, some Figures have also been remodeled based on the specific comments reported below.

  1. Some language in the description of results was a little vague, and should be clarified, either in legends or main text.

Introduction:

  1. Line 40: Good to change to severe acute respiratory syndrome 2 (SARS-CoV-2) as an initial definition for clarity

REPLY amended Line 40

  1. Line 45: “unequally distributed and worldwide available” – best to cite and clarify here, maybe “available worldwide, but distributed unequally”?

REPLY: amended Lines 45-46

  1. Lines 52-53: Could mention the receptor binding domain here – critical for binding to ACE2. Also need citation for function of ACE2.

REPLY: We rephrased the sentence and added a new reference (N^3 in the revised version) Lines 51-53  

  1. Lines 53-55: CRS citation needed – “ample evidence” but no attached references

REPLY: We added references (now N^ 4 and 5 corresponding to the N^14 and 15 of the previous version) Line 55

  1. Line 61: First line, it can be clearer that you’re talking about N-linked glycans, potentially with some background on their synthesis? O-linked glycans could be mentioned, but maybe not required.

REPLY: We replaced “saccharide constituent” with “N-linked glycans” Line 61

  1. Line 70-71: “that was” after the comma a little unclear – do you mean “which was”?

 REPLY: amended, line 70

  1. Line 81: Was there a particular baculovirus cell line that was used for S-in production? Would be helpful to mention alongside hek293 at this stage.

REPLY: Unfortunately, we don't have a precise information on this. In the data sheet of S-in protein we used (#40589-V08B1 Sino Biological) it is simply indicated that Expression host is Baculovirus-Insect cells. The insect source of the cell line could possibly be the moth Spodoptera frugiperda, whose cells are widely used as host of the baculovirus expression system.

Methods:

  1. Lines 91-93: Why would spike protein produced in baculovirus have an affinity (or ELISA binding? EC50 not quite clear on its own) of 400-1200, but spike protein from hek293 has a value ~40x higher (10-50)? Is ACE2 binding dependent on glycan structure?

REPLY: We do not have data to establish whether the differences in binding human ACE2 between S-in and S-hu of Sino Biologicals are dependent on their different glycan structures. For greater clarity, we have indicated that the EC50 refers to the binding ability in a functional ELISA. However, it has been reported that the site-specific N-linked microheterogeneity of Sars-Cov-2 Spike protein modulates spike–ACE2 interactions through the ability of these glycans to sterically mask polypeptide epitopes (Zhao, P., Praissman, J. L., Grant, O. C., Cai, Y., Xiao, T., Rosenbalm, K. E., et al. (2020). Virus-receptor interactions of glycosylated SARS-CoV-2 spike and human ACE2 receptor, Cell Host Microb. 28 (4), 586–601.e6. doi:10.1016/j.chom.2020.08.004). Lines 90-101. This said, our data would be in line with the suggestion that the N-glycans decorating the S-protein have an indirect role in the S-protein interaction with ACE2, possibly through S-protein stabilization (see Beyond Shielding: The Roles of Glycans in SARS-CoV-2 Spike Protein by Casalino L et.al BioRxiv preprint doi: https://doi.org/10.1101/2020.06.11.146522).

  1. Line 109: ELISA had spike protein coated at 0.75 ug/ml – was the concentration taken from the literature, or did it happen to work best for your assay? Would help to briefly address here.

 REPLY: After testing different concentrations, we selected 0.75 mg/ml because it was the lowest concentration useful for our experimental purposes.

  1. Lines 134-137: Says “the following stimulus” and lists three different stimuli – are these separate stimulus experiments, or are all items needed for appropriate stimulus? A little more description here would help.

 REPLY: The three stimuli were given independently of each other. The impact of S proteins, and in some cases of mannan, on these stimulations was then evaluated. We rephrased the sentences to make it clear. Lines 139-147.

  1. Line 141: Speed and duration for extraction of supernatant?

REPLY: We rephrased the sentence and added the missed information. Lines 147-150.

  1. Lines 177-179: Reference manufacturer of cytometer and developer of software

REPLY: BD Immunocytometer Systes, San Jose, CA was the manufacturer and the developer of software as indicated in the previous paragraph Lines 174-175.

Results:

  1. Line 191: Can focus on N-linked glycan moieties here, unless the references are talking about all kinds of glycans.

REPLY: the references indicating, in general, the importance of protein glycosylation in host-pathogen interaction, and do not exclusively focus on N-linked glycan moieties

  1. Figure 1A: Binding to Con A and inhibition of Con A binding by several molecules is shown, but only for S-in. Was the same analysis conducted with S-hu? It should be included to initially demonstrate binding, as is shown with S-in.

REPLY: According to the reviewer's suggestion, the data with S-hu have now added in Figure 1 A. Moreover, we introduced a Supplementary Figure S1 showing the percent inhibition of several scalar concentrations of mannan or alpha-methyl-mannoside on the binding of Con A to S-in or S-hu. The text in the results section was modified accordingly. Lines 204-212

  1. Figure 1A and 1B: The ug/ml units are a little hard to read in both figures – or does it say mcg/ml?

REPLY we have raised the font size in the Figure to make the wording more readable. The values are expressed in mg/ml

  1. Figure 1B: Surprised that S1 and S2 subunits bind to Con A at virtually the same level? Or maybe not, if the number of glycans on S1 and S2 is quite similar? Also, were these proteins produced in human or insect cells? Lastly, the S-hu, S1, and S2 portions of this panel have “+” for mannan and the protein of interest underneath every bar, which does not follow the pattern of +/- mannan in the S-in bars.

REPLY: compared to the trimeric form (S1+S2) of S-in or S-hu, both monomeric subunits of S-hu, S1 and S2, bound to HRP-Con A with lower efficiency. We don't know if this was due to being in monomeric form or the reduced number of glycans, particularly for the S2 subunit (nine N-glycans compared to 13 in the S1 subunit). We have corrected typos in the wording below the x-axis.

  1. Line 233: It says “Figure 1 A and B” as a reference, but the authors might mean Figure 2 A and B instead.

REPLY: the typo was amended line 246

  1. Lines 239-240: In some cases, the OD from the XTT assay was comparable, but in some cases the differences are significant. Can you comment on why cell viability might be significantly different between these conditions?

 REPLY: The OD values of the XTT test measure the activity of cells via mitochondrial dehydrogenases and depend on the number and metabolic activity of living cells. In the condition of CON A stimulation, the higher OD values were due both to the higher number of living cells and perhaps changes in cellular metabolism since CON A induced both cell proliferation and activation. For clarity, the first sentence was added in the M&M. Lines163-165 

  1. Figure 2A and 2B – overall the figures are good, but too small. These panels should be bigger to see all details clearly. Some elements of the panels are also unclear. It looks like there’s a dark blue line in each plot separate from the noted red and light blue lines. If kept, the significance of this line should be noted in the legend. Also, lines are connecting data points between groups, but it’s not clearly explained why. Are these lines indicating they are from the same donor? Any clarification in the legend would be helpful.

 REPLY: We enlarged some panels to make all the elements more readable. In the violin plot (panels A, B, E) the two dark lines represent the first and third quartiles. The individual plots inside violin plots or bars represents the individual donors and the lines connecting data points are from the same donor. The missed information has been added in the figure legends. Lines 275, 277-279.

  1. Figure 2C and 2E – the previous figures represent individual measurements as dots within the plot, but the same is not done for replicate data (three for 2C, six for 2E?) in these figures. It would be helpful to include the same information here. Also, the lines in the violin plots of 2E are likely referring to the same metrics as 2A and 2B (red line = median, etc.), but it is not stated in the legend explicitly.

 REPLY: We have added the individual plot also for figures in panel 2C and 2E, and the figure legends were rephased accordingly. See also our reply to the previous point. Lines 275, 277-279

  1. Figure 3A, B, D – same comment about the dark blue/black lines in violin plot – unclear of its significance.

REPLY: We explained the significance of black lines in this figure and also in all other figures with violin plot. Lines 316-318. See also our reply to point 7.

  1. Figure 3C – was the XTT assay used to measure cell viability for these images, and were any differences found?

REPLY: we added a new panel with data relative to XTT assay. The text in the result section Lines 294-297 and the figure legend were accordingly modified. Lines 310-313

  1. Line 304: The phrase “monocytes/macrophages” is a little unclear. Do you mean that both monocytes and macrophages are required, or that monocytes differentiating into macrophages are required?

      REPLY: we rephrased the sentence this way “soluble anti-CD3 mAb without agonist anti-CD28 mAb, as in our conditions, required signals from monocytes for fully T cell activation. Lines 329

  1. Lines 323-324/Figure 3D: The legend states that donors were assayed in duplicate, but that the data are plot from a run of “triplicate or quadruplicate”. More clarification on how data was obtained vs. how it is plotted would be helpful.

 REPLY: the incorrect sentence “Data are plot as mean ± sd of one experiment run in triplicate or quadruplicate” has been deleted Line 351

  1. Figure 5B: In the violin plot on the left, the median IL-6 protein level of donors with only MP65 stimulation is nearly three times higher than the median levels of MP65 alone in the other two panels. Was there anything different about the stimulation of these donors that would make the median so high compared to other groups? What range of measured IL-6 protein level might be expected in this assay following stimulation?

REPLY: Many more donors were used in S-in stimulation, some highly responsive to MP-65. Despite differences in the magnitude of IL-6 production in response to MP-65, neither S-in nor S-hu, as well as the mannan, do significantly impact MP65-mediated cytokine production. For clarity, we rephrased the sentence in the result section. Lines 359-362

  1. Lines 358-360: If the data will not be shown for this point, it would be helpful to add more details about these particular IFN gamma experiments. For instance, “even late in the culture” is vague about the timepoint(s) where stimulation did not occur. I would encourage the authors to include violin plots for this data in the same fashion as other panels in this manuscript. Even showing results with no significant differences would be helpful for transparency, and could be put in supplemental if allowed by the journal.

REPLY: We did not add violin plots because there is no production of IFN-g either in the unstimulated cells under medium culture condition or in cells stimulated with S-proteins or mannan. We have rewritten some sentences in the Result section to better explain these results and the number of times they were evaluated. Lines 387-389

  1. Line 365: The authors mention “late IL-6 accumulation,” but it’s not clear what late means in this context (at 96 hrs? before 96 hrs?). If applicable, referencing a specific timepoint in the assay would be more informative.

REPLY: We agree with the reviewer that the word “late” is confusing, so we deleted it. We meant that the IL-6 accumulation was measured at 96 h. Line 390

  1. Line 369: “Some morphological changes,” but no details are given by the authors. What morphological changes were observed, and how do they compare to previous panels under other stimulation conditions?

 REPLY: The morphological changes referred to higher numbers and size of cell agglomerates in cell cultures.  Of course, these changes were weaker compared to the agglomerates and clusters of activation found in cells stimulated with Con A (Figure 2 D and 3 C) or anti-CD3 (figure 4 C). We rephrased the sentence explaining the significance of morphological changes. Lines 398-399.

  1. Line 384: The red line showing the median values is mentioned, but the legend does not mention or describe the blue lines shown in violin plots of Figure 6B and 6C.

REPLY: We added the meaning of the two black lines in the violin plots. Instead, the light blue lines were not necessary in these violin plot, as the values of unstimulated cells grown only in medium are already part of the plot. Lines 416-417

  1. Lines 409-411: Suggesting that S-in has a higher affinity or avidity to CD206, but do not show direct evidence of binding. Can this be tested directly with CD206? Also, it is unclear whether the authors suspect direct contact of CD206 with glycans on spike, boosting of expression through some indirect mechanism, or both.

REPLY: We agree with the reviewer that we have no affinity/avidity data. However, considering that the MR continuously recycles between the inner cell surface and the cytoplasm and evaluating the effects at 30 min, we assume that the up-regulation of CD206 expression is, at least in part if not totally, due to the ability of the glycans of S proteins to interact with the mannose receptor. This interaction capacity appears to be greater for S-in than for S-hu. For the 72 h stimulation, however, the upregulation effect could be mediated by indirect mechanisms, too. We rephrased the sentences in the Results section, Lines 437-444 and in the Discussion section, Line 570.

  1. Figure 7A: A summary of CD206 positive cells with and without S-in is shown with violin plots, but a representative plot from the flow cytometer showing any gating, staining level, etc. (similar to Figure 2C) is not included. As with other pieces of data in this manuscript, it would help to show a representative image of flow CD206 positive % cytometry as well as the summary for transparency. Also, was this same experiment conducted with S-hu, and can those results be shown too?

REPLY: We agree and we added representative dot plots in Figure 7 A. We also included data for S-hu and mannan. Lines 452-453, 455, 457-458

  1. Figure 7B: The MFI of S-in appears to be a different level than the S-in MFI from Figure 7A. Is that the result of showing the mean instead of the median, or were different donors tested? A few more comments about this figure. First, the text of the axes for the panels showing counts and CD206 APC is quite small, and increasing the size would improve readability. Second, the histogram panel on the left does not indicate whether comparisons between the S-in, S-hu, and mannan groups are significant or not, as is shown in the panel on the right. Finally, the authors directly compare MFI of mannan with or without S-in or S-hu, but not the reverse. Can you test for significance there too (e.g. S-in in the left plot vs. mannan+S-in in the right plot)? If making those comparisons is possible, these panels could then be combined as well, making the figure a little less crowded.

REPLY. In panel 7 B of the old version, we have assayed 3 donors instead of 5 of plot in Figure 7A, all plots showing the mean of MIF.  We agree with the reviewer and we combine the two bar histograms in panel 7 B showing now the percentage of positive cells. We also show the significance among groups.

  1. Figure 7C: Same comment as Figure 7A – a representative cytometry plot could help, even in supplemental.

REPLY: we now show representative dot plots for CD206 expression on CD14+ macrophages in Figure 7(/2 h of culture) and Supplementary Figure S2 (last 30 min of culture). In the Supplementary Figure we also show the gating strategy of our analysis.

  1. Figure 7D: Tests for significance not shown among the S-in, S-hu, and mannan groups.

REPLY: Although only S-in significantly up-regulated CD206 expression, we found no significant differences between S-in and S-hu or mannan. All this perhaps due to the small number of samples analyzed.

Discussion:

  1. Lines 463-465: “A number of authors” are mentioned to support this sentence, yet no citations to specific studies are given.

REPLY: We have inserted a new reference (N^ 27) in addition to one already present (N^25 of previous version)

  1. Lines 466-467: The authors mention differences between glycan structures as a possible reason for differences in S-hu and S-in immunomodulation. Though exploring this hypothesis appears outside of the scope of this manuscript, this section would be more valuable if the authors discussed ways in which the effect of glycan structure could be directly investigated. For instance, have they thought about modifying glycans directly (e.g. stripping them off of glycosylated spike with an enzyme) or actively changing the composition of the glycans with GnTi- cells? Perhaps direct affinity measurements with CD206? I would encourage the authors to add their take on possible future directions.

 REPLY: We agree with the reviewer and in the Discussion section we have added the following sentence: “To resolve this relevant question, we intend to assay in the experimental models here described the S protein produced in E. coli, and therefore not glycosylated, and S-in or S-hu proteins with glycans modified by enzymatic treatments”. Lines 505-510

  1. Lines 484-485: Reference on Con A interaction with toll like receptors would be helpful here.

      REPLY: We added new references (N^ 31 and 32) in addition with the N^ 34 of the previous version, now N^33.  Line 528

  1. Line 491: “The sugars” – do you mean N-glycans specifically? If so, it would help to note here and include a citation for the sentence.

REPLY: We replaced “sugars” with “glycoproteins carrying N- and O-linked oligosaccharides” and added the reference (now N^ 34 corresponding to N^ 29 of the previous version). Lines 532, 533, 535.

  1. Lines 507-512: The authors make an interesting point about potential interactions between vaccination with spike antigens and existing immunomodulatory therapies. Could this potential effect extend beyond COVID (thinking especially of immunization with influenza and RSV glycoproteins), or does spike protein have unique immunomodulatory effects that make this a concern? Commenting on this possibility could be informative and broaden the scope of suggested future directions.

REPLY: This is a very interesting point and we know there is very active research, in this context, for both the viruses indicated by the reviewer as well as for HIV glycoproteins experimentally tested for vaccination. However, we don’t have, at the moment, a foot in the updated research with viruses other than SARS-CoV-2 fields, we would like to avoid other discussions or commenting here.

  1. Lines 513-517: First two sentences of this paragraph introduce some points about CD206, but do not have citations.

REPLY: we added new references (N^ 41 and 42) Lines 557, 559.
